# Dissecting Molecular Heterogeneity of Circulating Tumor Cells (CTCs) from Metastatic Breast Cancer Patients through Copy Number Aberration (CNA) and Single Nucleotide Variant (SNV) Single Cell Analysis

**DOI:** 10.3390/cancers14163925

**Published:** 2022-08-14

**Authors:** Tania Rossi, Davide Angeli, Michela Tebaldi, Pietro Fici, Elisabetta Rossi, Andrea Rocca, Michela Palleschi, Roberta Maltoni, Giovanni Martinelli, Francesco Fabbri, Giulia Gallerani

**Affiliations:** 1Biosciences Laboratory, IRCCS Istituto Romagnolo per lo Studio dei Tumori (IRST) “Dino Amadori”, 47014 Meldola, Italy; 2Unit of Biostatistics and Clinical Trials, IRCCS Istituto Romagnolo per lo Studio dei Tumori (IRST) “Dino Amadori”, 47014 Meldola, Italy; 3Department of Surgery, Oncology and Gastroenterology, University of Padova, 35124 Padova, Italy; 4Veneto Institute of Oncology IOV-IRCCS, 35128 Padua, Italy; 5Department of Medical Oncology, IRCCS Istituto Romagnolo per lo Studio dei Tumori (IRST) “Dino Amadori”, 47014 Meldola, Italy; 6Healthcare Administration, IRCCS Istituto Romagnolo per lo Studio dei Tumori (IRST) “Dino Amadori”, 47014 Meldola, Italy; 7IRCCS Istituto Romagnolo per lo Studio dei Tumori (IRST) “Dino Amadori”, 47014 Meldola, Italy

**Keywords:** metastatic breast cancer, circulating tumor cells, copy number aberrations, single nucleotide variants, single cell sequencing, next generation sequencing, liquid biopsy

## Abstract

**Simple Summary:**

Circulating tumor cells (CTCs) are rare cells found in the bloodstream of oncologic patients with a central role in the metastatic spread. In this study, we aim at exploring their heterogeneity levels in metastatic breast cancer patients focusing on single cell single nucleotide variant (SNV) and copy number aberration (CNA) analyses. Our results show high levels of heterogeneity, especially concerning SNVs. Further analysis revealed the presence of CNAs associated with breast tumorigenesis, while longitudinal CNA profiling was demonstrated to track clonal selection of CTCs during treatment. Despite this heterogeneity, we found a group of CTCs from different patients sharing common genomic aberrations, such as losses on 15q. Collectively, our findings demonstrate that single-cell molecular analyses could be exploited in future to better address therapeutic strategies. Further investigations to better characterize this mixed population are needed to understand its role in MBC.

**Abstract:**

Circulating tumor cells’ (CTCs) heterogeneity contributes to counteract their introduction in clinical practice. Through single-cell sequencing we aim at exploring CTC heterogeneity in metastatic breast cancer (MBC) patients. Single CTCs were isolated using DEPArray NxT. After whole genome amplification, libraries were prepared for copy number aberration (CNA) and single nucleotide variant (SNV) analysis and sequenced using Ion GeneStudio S5 and Illumina MiSeq, respectively. CTCs demonstrate distinctive mutational signatures but retain molecular traces of their common origin. CNA profiling identifies frequent aberrations involving critical genes in pathogenesis: gains of 1q (*CCND1*) and 11q (*WNT3A*), loss of 22q (*CHEK2*). The longitudinal single-CTC analysis allows tracking of clonal selection and the emergence of resistance-associated aberrations, such as gain of a region in 12q (*CDK4*). A group composed of CTCs from different patients sharing common traits emerges. Further analyses identify losses of 15q and enrichment of terms associated with pseudopodium formation as frequent and exclusive events. CTCs from MBC patients are heterogeneous, especially concerning their mutational status. The single-cell analysis allows the identification of aberrations associated with resistance, and is a candidate tool to better address treatment strategy. The translational significance of the group populated by similar CTCs should be elucidated.

## 1. Introduction

Recent updates concerning cancer statistics have reported that, in 2020, breast cancer (BC) ranked as the most commonly diagnosed tumor type worldwide [1]. Despite the high morbidity of BC, survival at 5 and 10 years has higher rates compared to other cancer types, especially in developed countries, where massive screening is successfully applied to detect BC at early stages [2]. However, these rates dramatically drop for patients diagnosed with metastatic BC (MBC), which still represents an incurable disease [3].

Circulating tumor cells (CTCs) consist of a rare population of cells that detach from the tumor mass towards secondary organs, thus being among the main perpetrators of the metastatic spread of epithelial tumors [4]. In MBC, CTC enumeration was proven to be an independent prognostic factor of progression-free survival (PFS) and overall survival (OS), displaying poor prognosis in patients with basal CTC count ≥5 in 7.5 mL of blood [5]. Furthermore, baseline enumeration of CTCs was shown to predict the progression of disease earlier than the standard timing of anatomical assessment using conventional radiological tests in BC patients with limited metastatic dissemination [6]. In recent years, the molecular heterogeneity of CTCs has been deeply investigated, and intra-patient heterogeneity is considered nowadays a matter of fact with manifest clinical implications [7]. Up to now, next-generation sequencing (NGS) is the master technique for the identification of alterations associated with pathogenesis [8,9,10]. Hence, molecular characterization of CTCs through NGS could provide critical information concerning the primary tumor in MBC and represents a fascinating tool to improve precision medicine [11].

In this study, we aimed at exploring the heterogeneity of CTCs from MBC patients through single-cell single nucleotide variant (SNV) and copy number aberration (CNA) analysis. In brief, our findings highlight great levels of intra- and inter-patient CTC heterogeneity, especially concerning their SNVs. Furthermore, the herein reported detection of aberrations involving clinically-relevant genes, even over time, supports the role of CTCs as a “liquid biopsy” to better address therapies in the near future. Finally, we identified a group populated by similar CTCs from different patients that need further analysis to delineate their role in MBC pathogenesis.

## 2. Materials and Methods

### 2.1. Patients

Six patients diagnosed with metastatic luminal (estrogen receptor (ER) and progesterone receptor (PgR) positive, Human Epidermal Growth Factor Receptor 2 (HER2) negative) BC were enrolled in this study. Blood withdraws were performed at three timepoints: at baseline (timepoint A), 6–8 weeks after the beginning of treatment (timepoint B) and 2–4 weeks after the last dose of therapy (timepoint C). All subjects gave written informed consent to the conservation and use of the samples for research purposes. The study was conducted in accordance with the Declaration of Helsinki, and the protocol was approved on 9 November 2018 by the Romagna Ethics Committee (CEROM) of Meldola (IRST174.19).

### 2.2. CTC Enrichment, Recovery and Whole Genome Amplification (WGA)

CTCs and white blood cells prestained with antibodies to CD45 and pan-CK and DAPI were aspirated from the CellSearch cartridge used for the CTC enumeration, and single cells were isolated using DEPArray NxT (Menarini-Silicon Biosystems SpA, Bologna, Italy). After isolation of cells in PCR-sterile tubes, we performed volume reduction and 1X PBS wash under a sterile hood. Samples were stocked at −20 °C until further analysis.

To obtain evaluable DNA for downstream analysis, we performed WGA using Ampli1 WGA kit (Menarini-Silicon Biosystems SpA, Bologna, Italy) and the quality of amplified DNA was assessed using the Ampli1 QC kit (Menarini-Silicon Biosystems SpA, Bologna, Italy). The PCR product was evaluated through electrophoresis on a 2% agarose gel and visualized using Chemidoc XRS System (Bio-rad, Hercules, CA, USA). Based on the number of visualizable bands for each sample we proceeded with library preparation. WGA products were conserved at −20 °C until further downstream analysis.

### 2.3. Library Preparation and Sequencing

For CNA analysis, libraries were prepared from samples with at least two out of four bands at the quality control. The preparation of the libraries was performed using the Ampli1 Lowpass kit for Ion Torrent (Menarini-Silicon Biosystems SpA, Bologna, Italy) following the protocol provided by the manufacturer. Final library concentration and quality were assessed by Qubit 4.0 Fluorometer (Thermo Fisher, Waltham, MA, USA) and Bioanalyzer High Sensitivity DNA (Agilent Technologies, Waldbronn, Germany), respectively. Equimolar pools were prepared and loaded into Ion 520 Chips using the Ion Chef System (Thermo Fisher, Waltham, MA, USA). Sequencing was carried out using the Ion Torrent S5 System.

Samples with at least three out of four bands at the quality control were considered eligible for SNV analysis. Libraries were prepared using the Ampli1 OncoSeek Panel (Menarini-Silicon Biosystems SpA, Bologna, Italy) and quantified using the KAPA Library quantification Kit (Roche, Basel, Switzerland). Equimolar pools were loaded on 300-cycles V2 cartridges and sequenced 2 × 151 on MiSeq sequencing system (Illumina Inc., San Diego, CA, USA).

### 2.4. Bioinformatic and Statistical Analyses

The bioinformatics analyses were performed with customized pipelines. For Ampli1 OncoSeek Panel, raw de-multiplexed reads from the MiSeq sequencer were trimmed of primer sequences and successively aligned to the reference human genome (UCSC-Build37/hg19) using the Burrows–Wheeler algorithm [12], running in paired-end mode. To ensure good call quality and to reduce the number of false positives, samples underwent Base Quality Score Recalibration (BQSR), using the Genome Analysis Toolkit GATK, version 3.2.2 [13]. After BQSR, sequences around regions with insertions and deletions (indels) were realigned locally with GATK. For somatic variant analysis VarScan 2.3.9 [14] was used to search for SNVs and indels. Genomic and functional annotations of detected variants were made by Annovar [15]. Coverage statistics were performed by DepthOfCoverage utility of GATK. BASH and R custom scripts.

For CNA calling, Control-FREEC [16] was used starting from sorted bam files derived by Ion Torrent S5 System and samtools [17], and a quality check of our samples was performed by computing the derivative log ratio spread (DLRS) [18]. Then, we applied two non-parametric tests (Mann–Withney and Kolmogorov-Smirnov) computing *p*-values in order to assess the statistical significance of each call and filtered out preliminary CNAs having at least one of the two *p*-values higher than 0.05. From each CNA, we extracted the list of genes spanning the aberration and performed enrichment analysis on Gene Ontology (GO) datasets [19] for each sample with EnrichR [20]. Next, we applied the Genomic Identification of Significant Targets in Cancer (GISTIC) tool [21] in order to identify regions with significant aberrations inside a group of samples. Finally, we performed hierarchical clustering on different groups of samples based on copy numbers inside bins of 1 million bases, computing distance among samples with Euclidean metrics.

## 3. Results

### 3.1. CTC Isolation in MBC Patients and Quality Assessment

Epithelial CTCs were enriched using the CellSearch system, and single-cell isolation was performed using the DEPArrayNxT platform. Overall, we isolated 124 single pure CTCs from 6 MBC patients (Table 1).

At QC check, 35 CTCs (28.23%) displayed 4 PCR bands, whereas 24 (19.35%) exhibited 3 bands. Twenty-eight (22.58%) showed medium quality with 2 out of 4 bands. Thirty-seven CTCs were considered not suitable for downstream analysis: 16 (12.90%) and 21 CTCs (16.94%) showed, respectively, 1 and 0 bands. An example of the results obtained after PCR products run on agarose gel electrophoresis is reported in Appendix A. For each patient, we isolated one 10-lymphocyte pool for normalization in downstream analysis. All the lymphocyte pools demonstrated optimal quality for downstream analysis, displaying 4/4 bands at the PCR-based quality control check.

### 3.2. Single Nucleotide Variant (SNV) Analysis on Single CTCs

To gain information concerning the somatic molecular features of CTCs, 59 CTCs from 4 patients whose amplified DNA showed appropriate quality (at least 3 PCR bands) were tested for a hotspot panel of 60 tumor-related genes. Additional analysis conducted of lymphocyte-pool samples excluded the presence of candidate germinal mutations in all the patients. Overall, called variants were observed in 32 out of 55 evaluable CTCs (58.2%). Regarding CTCs from patient CH28, we obtained poor sequencing quality in terms of coverage statistics, thus the results are unavailable. CTCs from patients CH46 and CH47 were not included in the SNV analysis having less than 3 bands at QC. The list of the alterations found in single CTCs is reported in Appendix A.

In patient CH29, variants were called in 15/17 (88%) CTCs. We found three shared variants with unknown clinical significance: *ATM* S333F (12/15 CTCs; 80%) persistent at all the timepoints, *TP53* R210X (3/15 CTCs; 20%) from timepoints A and B, *PTEN* S287X (2/15 CTCs; 13%) from timepoint B. At time C, we observed in a single CTC the RB1 F755S variant. Despite its clinical significance is still unknown, somatic mutations affecting this locus have been reported as an acquired mechanism of CDK4/6 inhibitors resistance [22].

Concerning patient CH30, while 10 CTCs did not display any alteration affecting the tested genes, we found exclusive exonic variants in 13 out of 23 CTCs (56.5%). Notably, two mutations with clinical significance were observed: the nonsynonymous SNVs *PIK3CA* N345D [23] (1 CTC from timepoint A) and *MLH1* S127L [24] (rs201673334; 1 CTC from timepoint A).

Patient CH32 displayed 4 out of 15 CTCs (26.67%) from timepoint A with called variants. We found in 2 CTCs (50%) the nonsynonymous SNV *AKT1* E17K (rs121434592), reported as pathogenic in several cancer types including BC and associated with resistance to therapies [25,26,27]. In addition, two mutations affecting the *TP53* locus were identified: a frameshift deletion occurring on the DNA binding domain *TP53* S128fs (2/4 CTCs; 50%) and the nonsynonymous SNV *TP53* V25F (rs121912654, 1/4 CTC; 25%).

In summary, CTCs harbor alterations associated with pathogenesis and therapy resistance, and the high intra- and inter-patient heterogeneity levels make the translational interpretation of identified variants in CTCs challenging.

### 3.3. Single Cell Profiling of CTCs

Considering SNV results, we aimed at further exploring the molecular characteristics of CTCs by profiling their CNAs.

A total of 87 single CTCs were sequenced, and the quality was checked by computing the DLRS [18]. 77/87 samples (88.15%) were considered suitable for further bioinformatic analyses (DLRS values < 0.3). Data of CTCs from patient CH28 are unavailable due to quality issues, having a DLRS ≥ 0.3, and the good quality of copy number profile of the leukocyte pools of patient CH28 excluded procedure shortcomings (Appendix A). The lack of significative aberrations on lymphocyte pools allowed us to confirm the tumoral nature of identified CNAs. The CNA profile plots of representative CTC are reported in Appendix A.

To investigate the frequency and the amplitude of segmented copy number values, we used the GISTIC tool. The plots in Figure 1 report the frequent gains and losses that occurred in all the CTCs (n = 77).

Overall, this analysis identified frequent amplifications on chromosomes 1q, 4p, 11q, 14q, 16p and 22q. The most frequent amplified region was a 183.70 kb long region on chromosome 22q11.2, containing only the *KIAA1671* gene whose product was previously considered as a candidate biomarker for early detection of in situ BC [28]. Of relevance, the gain of 11q24.2 involves the *CCND1* gene [29].

Among the most frequent deletions, we found loss on chromosome 13q. Notably, this region hosts the *RB1* locus, codifying a crucial tumor-suppressor whose loss is associated with resistance to anti-CDK4/6 therapies in ER-positive BC [28]. In addition, we found the deletion of the tumor suppressor gene *CHEK2* in chromosome 22q13.3, a region commonly found loss in BC [30].

Taken together, data from GISTIC analyses identify the presence of aberrations in CTCs associated with breast pathogenesis.

### 3.4. Longitudinal Investigation of CTC CNA Profiles

Next, we aimed at identifying patient-specific aberrations in CTCs involved in tumor progression and CTC survival. To this purpose, we applied the GISTIC algorithm to CTCs of patients CH29 and CH30, having CTCs from at least two time points.

Patient CH29 (treated with fulvestrant plus palbociclib) displayed heterogeneous aberration spectra across the time points and gain of chromosome 16q12.2 emerged as a persistent alteration. Gains in regions 1q41 and 12q12 emerged as frequently in gain in time B and persistence in time C, suggesting their occurrence as a result of selection pressure exerted by therapy. Indeed, region 12q12 hosts the *CDK4* gene whose overexpression represents one of the main mechanisms of resistance to CDK4/6 inhibitors [31] (Figure 2).

Concerning patient CH30 (treated with vinorelbine plus capecitabine), frequent aberrations in CTCs from timepoints A and C were gain of chromosome 11q13.2 and chromosome 22q11.2 (Figure 3).

The region 11q13 is a gene-rich region found amplified in 15% of all primary breast tumors [32], and hosts genes known as potential contributors to positive selection for cell proliferation and survival by previous studies, among which the oncogene *CCND1* [33].

Collectively, the liquid biopsy of two differently treated MBC patients confirmed CTC heterogeneity in terms of CNA profiling, while some alterations persist along the course of treatment and host genes associated with resistance to anti-tumor agents.

### 3.5. Identification of a Population of CTCs with Common Traits

Next, in order to better understand the levels of CTC heterogeneity, we integrated single-cell data from all the patients. For this purpose, we performed principal component analysis (PCA) for dimensionality reduction and unsupervised visualization of CNA data (Figure 4a).

Based on our results, single CTCs from the same patient have the tendency to group together, although a subpopulation of mixed CTCs from different patients was present (crossed-patient CTC set). The existence of a crossed-patient CTC set was confirmed by the hierarchical clustering tree (Figure 4b). The crossed-patient CTC set was populated by 15 cells, and each patient had at least one CTC grouping within this set.

Next, we applied GISTIC 2.0 and term enrichment analyses to explore the molecular and functional features of the crossed-patient CTC set (Appendix A), and the results were compared with those obtained from the entire CTC case series using the same tools (Appendix A).

The most frequent private aberrations in the crossed-patient CTC set were gain in region 9q34.3 (813 kb) and a great loss borne by 15q (71.80 Mb), which is frequently reported in the literature in association with tumor progression and metastasis in BC [34,35,36,37] (Appendix A). In Table 2, we reported the genes included within 15q with a tumor-suppressor role in BC.

The results of the enrichment analyses showed that the most enriched term in GO Biological Process was “positive regulation of transforming growth factor beta production (GO:0071636)” in 14/15 CTCs (93.3%). However, this term is not considered privately enriched in the crossed-patient CTC set, as it was also found enriched less frequently in CTCs from the entire case series (51/77 CTCs; 66.2%). The “pseudopodium (GO:0031143)” (GO Cellular Component database) was another frequent enriched term in the crossed-patient CTC set (9/15 CTCs; 60%), while it was less enriched in the entire case series (18/77 CTCs; 23.4%).

Our data infer the existence of a crossed-patient CTC set with specific molecular and functional characteristics. Our findings sketch various levels of heterogeneity: (I) CTCs tend to group in a patient-specific manner, and (II) some CTCs are similar to each other while belonging to different patients. Hence, despite a context of evident heterogeneity, our data reveal the existence of a hidden similarity among CTCs.

## 4. Discussion

In recent years, CTCs have been deeply investigated due to their pivotal role in the metastatic cascade, but their high heterogeneity still contributes to counteract their introduction in clinical practice [7,53]. In fact, as a consequence of the dynamic changes in the bloodstream, CTCs can converge in distinct subpopulations with specific genomic and phenotypic features [54]. Up to now, CTC enumeration remains the only FDA-approved CTC-based assay in clinical practice with prognostic purpose [55], but the heterogeneous genetic features characterizing this cell population could no longer be neglected as they dramatically influence disease manifestation and outcome [56]. Hence, the identification of novel approaches to further exploit data from CTCs for translational purposes beyond CTC enumeration is imperative. The aim of this study is to explore the molecular landscape of CTCs in MBC patients. For this purpose, we exploited single-cell resolution NGS to assess CTC mutational status and CNA profile. However, this study has some flaws. Firstly, the limited number of patients makes challenging to achieve robust statistical data. However, this weakness is mitigated by high-quality single-cell data illustrated in this article, especially with whole-genome CNA profiling. Moreover, the availability of primary tumor and metastatic specimens would have been helpful to integrate the mutational profiling, and to better delineate the clonal selection of CTCs based on their CNAs.

Our results confirm that single-cell analysis of CTCs represents an accurate approach to unmasking genetic heterogeneity. In our study, testing single CTCs for a panel of 60 cancer-relevant genes revealed a great intra-patient degree of variant heterogeneity, supporting the existence of distinct self-specific mutational signatures for each CTC. Our study is not the first to describe the detection of mutational heterogeneity in single CTCs from MBC patients [11,57,58,59,60].

At the same time, while no mutations were common among patients, we detected a limited number of alterations shared among CTCs within the same patient even at different time points. This finding, as observed by other researchers [11], highlights that the presence of shared molecular alterations reflects the common clonal origin of tumor cells in the bloodstream [61]. Overall, due to the high degree of heterogeneity, the translational significance of candidate alterations is still not univocal. One key example is represented by the *AKT* E17K variant which is reported as pathogenic and associated with resistance to therapies in BC [62] but was retrieved in 50% of CTCs at timepoint A in patient CH32. Finally, we found a limited number of CTCs with called SNVs. This result may be imputable to the tested panel, which includes a limited number of genes and covers hotspot mutations. Hence, we cannot exclude the presence of alterations occurring on other genes not included in this analysis.

Concerning chromosomal aberration analysis, while confirming the heterogeneous portrait of CNAs in CTCs, we described the emergence of frequent alterations involving genes with a critical role in MBC in agreement with the literature. This workflow was previously successfully exploited by our group for the longitudinal characterization of single CTCs from early BC [7] and oesophageal cancer patients [63]. Herein, we found that gain of the region 22q11.2 emerged as the most frequent aberration in CTCs. This short aberration includes a unique gene, *KIAA1671*, which codifies for an uncharacterized protein involved in mitosis and chromosome segregation [64]. Although its role is still unexplored, *KIAA1671* seems to contribute to BC pathogenesis, since autoantibodies against the codified protein were detected in serum from BC patients [28]. Moreover, emerging frequent aberrations included several gains (1q, 4p, 11q, 14q and 16p) and losses (13q and 22q), among which aberrations already described in BC and often hosting clinically-relevant genes. One example is represented by *WNT3A* (1q42), whose protein product encompasses a wide range of roles, from oncogenesis to developmental processes. In vitro studies have demonstrated a role for *WNT3A* in BC proliferation [65] and resistance to tamoxifen treatment [66]. *CCND1* (11q13) amplification is frequent in BC and associated with progression and resistance [29,67]. The encoding product, Cyclin D1, is an oncogenic protein with a pivotal role in G1 to S phase transition during the cell cycle [33,68]. A high copy number of *CCND1* was previously found to identify a subset of ER-positive patients with poorer prognosis [69]. Beyond its role in cell cycle regulation, Cyclin D1 harbours also non-canonical cdk-independent functions, among which the ability to interact with the hormone binding domain of ER. As a consequence, ER-mediated transcripts are upregulated even in absence of the canonical ligand [70]. On the other hand, Checkpoint Kinase 2, codified by the *CHEK2* (22q12.1) gene, acts as a tumor-suppressor in BC where gene loss is not infrequent [71,72].

In addition, longitudinal analyses on single cells from MBC patients with CTCs in at least two timepoints allowed us to track the clonal selection of CTCs during treatment, supporting the power of CTC CNA profiling as an informative approach in patient monitoring [7,53,73]. For instance, being 16q12.2 in gain in all the timepoints in patient CH29, we consider this aberration as evidence of a resistant clone that escapes the selective pressure exerted by treatment with fulvestrant and palbociclib. The presence within this region of loci codifying for protein with a central role in EMT and cell migration, among which *AMFR* and genes encoding metallothioneins [74,75], gives a further demonstration. In addition, the occurrence of CTC clones harboring gain 12q12 clearly reflects the selection exerted by palbociclib, since the amplification of the *CDK4* gene represents one master mechanism of resistance of this agent [31]. Analogously, the presence of therapy-resistant clones was illustrated also in patient CH30, whose CTCs harbored gains in 11q13.2 and 22q11.2 before chemotherapy started, and persisted at the end of therapy. Again, critical genes are hosted within these regions, among which *CCND1*.

Last, besides marked inter-patient CTC heterogeneity, in-depth analyses revealed the emergence of a mixed group populated by CTCs with common genomic traits, being more similar to each other compared to other cells from the same patient. Interestingly, we noted that losses occurring on the long arm of chromosome 15 were frequent and appeared exclusive to the mixed group. Interestingly, copy number losses of 15q might have a profound effect on the expression of the multitude of tumor suppressor genes hosted. Therefore, the involvement of aberrations occurring in 15q in several cancer types, among which bladder [76], head-and-neck [36], and breast tumors [34,77], is not new. In addition, through enrichment analyses we found that the mixed population had the term associated with pseudopodium generation was enriched. Pseudopodia are actin-dependent cell protrusions implied in mesenchymal tumor cell migration, and inhibition of the protein involved has been proposed as a potential strategy for metastasis blockade [78,79]. To the best of our knowledge, this is the first description of a further mixed population, but understanding the exact significance of this crossed-patient CTC set is nearly impossible within our analyses. However, since the enrolled patients had undergone breast surgery before the diagnosis of metastasis, we cannot exclude that this condition could be derived from the different sources of CTCs, i.e., occult niches and metastasis. Further analyses on a higher number of cases and CTCs and on the metastatic site and primary tumor would be helpful to address this topic.

## 5. Conclusions

Through single cell analysis, we found that MBC CTCs display self-specific mutational signatures but might retain molecular traces of their clonal origin. Moreover, single-cell CNA profiling brings out the presence of frequent aberrations implying genes involved in breast tumorigenesis, while longitudinal analyses allow us to track the clonal selection of CTCs during treatment. Moreover, despite the great heterogeneity, a group of CTCs from different patients sharing common genomic traits emerged, displaying losses on 15q and enrichment of terms associated with pseudopodium generation as frequent events. Collectively, our findings demonstrate that single-cell molecular analyses could be exploited in the future to better address therapeutic strategies. Further investigations to better characterize this mixed population are needed to understand its role in MBC.

## Figures and Tables

**Figure 1 cancers-14-03925-f001:**
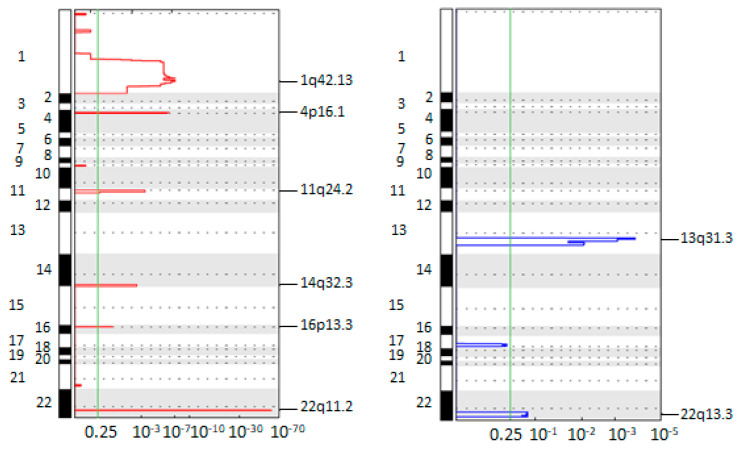
Genomic Identification of Significant Targets in Cancer (GISTIC) amplification (left, red) and deletion (right, blue) plots on single circulating tumor cells (CTCs) of all the patients. The genome is oriented vertically from top to bottom, and the GISTIC q-values at each locus are plotted from left to right on a log scale. The green line represents the significance threshold (*q*-value = 0.25).

**Figure 2 cancers-14-03925-f002:**
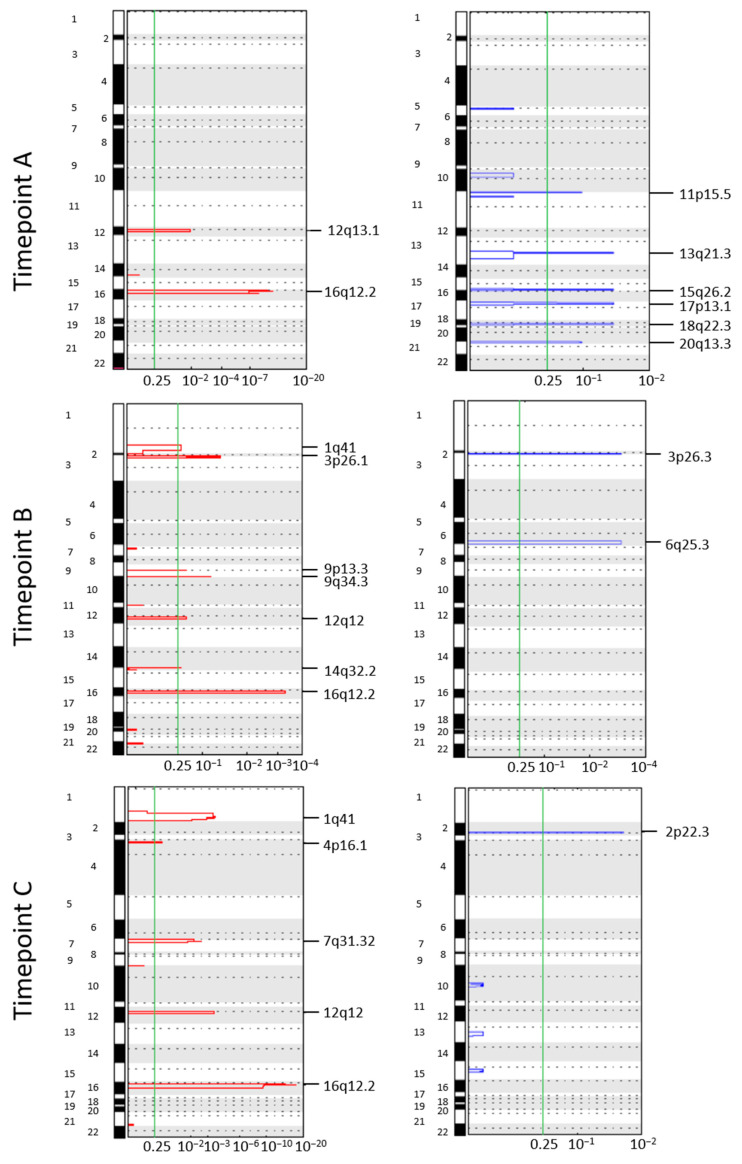
Genomic Identification of Significant Targets in Cancer (GISTIC) amplification and deletion plots on single circulating tumor cells (CTCs) from patient CH29. Each column represents a different timepoint. The genome is oriented vertically from top to bottom, and the GISTIC q-values at each locus are plotted from left to right on a log scale. The green line represents the significance threshold (*q*-value = 0.25). A: basal (10 CTCs); B: screening (6–8 weeks after; 5 CTCs); C: end of therapy (13 CTCs).

**Figure 3 cancers-14-03925-f003:**
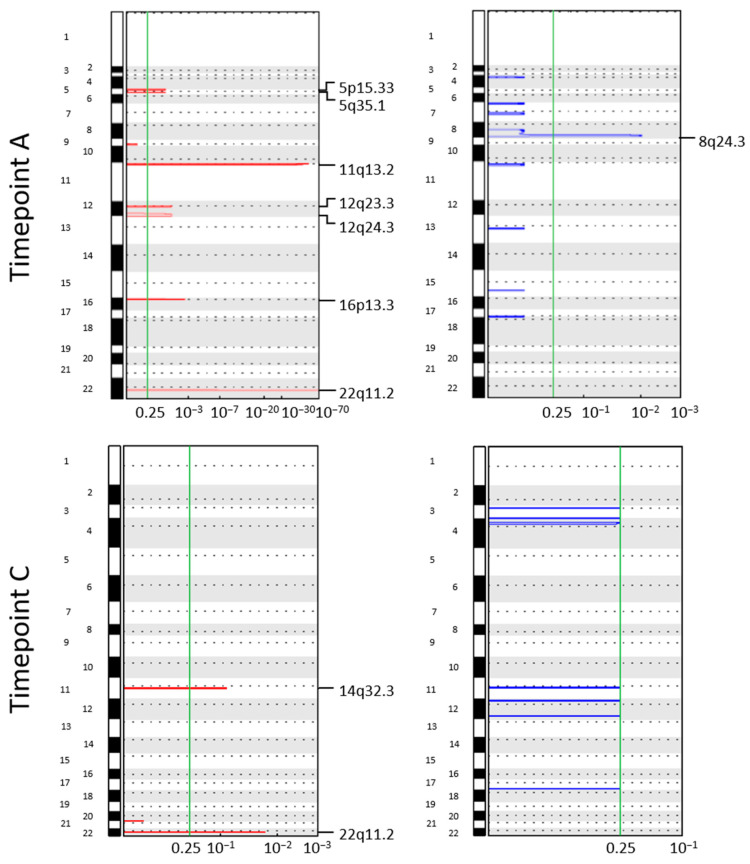
Genomic Identification of Significant Targets in Cancer (GISTIC) amplification and deletion plots on single circulating tumor cells (CTCs) from patient CH30. Each column represents a different timepoint. The genome is oriented vertically from top to bottom, and the GISTIC *q*-values at each locus are plotted from left to right on a log scale. The green line represents the significance threshold (*q*-value = 0.25). A: basal (37 CTCs); C: end of therapy (2 CTCs).

**Figure 4 cancers-14-03925-f004:**
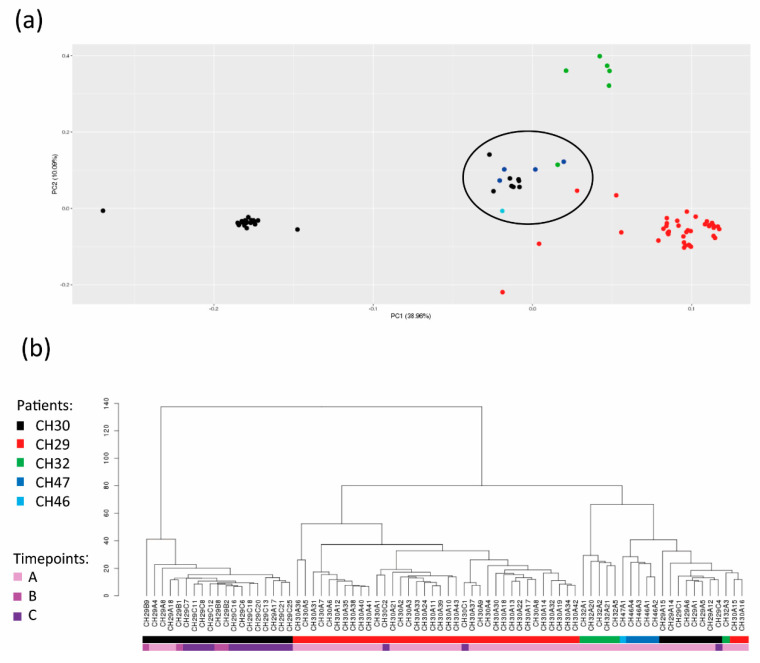
Intertumoral heterogeneity of circulating tumor cells (CTCs). (**a**) Principal component analysis (PCA) plot was generated using all the CTCs of all the patients. Each dot corresponds to a single CTC of each patient with a colour-based code. Black: patient CH29; Red: patient CH30; Green: patient CH32; Blue: patient CH46; Cyan: patient CH47. The black circle highlights the crossed-patient CTC set. (**b**) Hierarchical clustering tree.

**Table 1 cancers-14-03925-t001:** List of patients with clinical parameters and number of single CTCs for each patient and timepoint.

Patient ID	Therapy	Clinical Response	Site of Metastasis	Number of CTC Isolated			
				Timepoint A	Timepoint B	Timepoint C	Total
CH28	Capecitabine	PR	Bone, Lymph nodes	5			5
CH29	Fulvestrant + Palbociclib	PD	Peritoneum	17	9	30	56
CH30	Capecitabine + Vinorelbine	PD	Bone, Liver	37		2	39
CH32	Capecitabine + Vinorelbine	SD	Bone, Liver, Lung	15			15
CH46	Letrozole + Ribociclib	SD	Bone	5			5
CH47	Letrozole + Ribociclib	N/A	Bone, Lymph nodes	4			4
			Total	83	9	32	124

CTC: circulating tumor cell; PR: partial response; PD: progressive disease; SD: stable disease; N/A: not available.

**Table 2 cancers-14-03925-t002:** List of tumor suppressor genes comprised within the region 15q.

Gene	Gene Product	Genomic Location	References
*BLM*	Bloom Syndrome RecQ Like Helicase	15q26.1	[38]
*ONECUT1*	One Cut Homeobox 1	15q21.3	[39]
*PML*	Promyelocytic Leukemia Protein	15q22	[40]
*THBS1*	Thrombospondin-1	15q15	[41]
*TP53BP1*	Tumor Protein P53 Binding Protein 1	15q15-q21	[42]
*ANP32A*	Acidic Nuclear Phosphoprotein 32 Family Member A	15q23	[43]
*ALDH1A2*	Aldehyde Dehydrogenase 1 Family Member A2	15q21.3	[44]
*CCNDBP1*	Cyclin D1 Binding Protein 1	15q14-q15	[45]
*BMF*	Bcl2 Modifying Factor	15q14	[46]
*ST20*	Suppressor Of Tumorigenicity 20	15q25.1	[47]
*MIR211*	Hsa-Mir-211	15q13.3	[48]
*MIR7-2*	Hsa-Mir-7	15q26.1	[49]
*MIR9-3*	Hsa-Mir-9	15q26.1	[50,51]
*MIR422A*	Hsa-Mir-422a	15q22.31	[52]

## Data Availability

The datasets used and/or analyzed during the current study are available from the corresponding author on reasonable request.

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
