# Peer review of "Dissecting Molecular Heterogeneity of Circulating Tumor Cells (CTCs) from Metastatic Breast Cancer Patients through Copy Number Aberration (CNA) and Single Nucleotide Variant (SNV) Single Cell Analysis"

_cancers, 2022, doi:10.3390/cancers14163925_

Round 1

Reviewer 1 Report

This article explored the CTCs heterogeneity levels in metastatic breast cancer patients focusing on SNVs and CNAs analyses. They have found some SNVs and CNAs associated with pathogenesis and treatment resistance and identified a group of CTCs from different patients sharing common genomic aberrations. These results provide new insights into the study of MBC mechanisms.

Minor issue:

1. Only patient CH29 has relatively sufficient number of CTCs for analysis in all three time points. Patient CH30 only has 2 CTCs for time point C. Any analysis and interpretation based on these data cannot be considered to have enough statistical confidence. Results relating to the analyses on CTCs from patient CH30 on time point C should be moved from the main text into supplemental materials.

2. The caption of section 3.3 and 3.4 are the same. Please fix.

3. GO enrichment results should be provided in supplemental materials.

Author Response

We thank the reviewer for revising our manuscript and for the positive comments. We are confident that the suggestions significantly improved our manuscript. Please find below a point-by-point answer to the comments. 

This article explored the CTCs heterogeneity levels in metastatic breast cancer patients focusing on SNVs and CNAs analyses. They have found some SNVs and CNAs associated with pathogenesis and treatment resistance and identified a group of CTCs from different patients sharing common genomic aberrations. These results provide new insights into the study of MBC mechanisms.

Minor issue:

  1. Only patient CH29 has relatively sufficient number of CTCs for analysis in all three time points. Patient CH30 only has 2 CTCs for time point C. Any analysis and interpretation based on these data cannot be considered to have enough statistical confidence. Results relating to the analyses on CTCs from patient CH30 on time point C should be moved from the main text into supplemental materials.

Reply: we thank the reviewer for the kind suggestion. We are aware of the big difference of CTC number between patient CH29 and CH30. To highlight the number of cells investigated in this analysis, we added the CTC count in lines 275-276 and 284-285. Being CTCs extremely rare, longitudinal comparison even with a low count of cells is not infrequent. Hence, given the results, we prefer to maintain the analyses conducted on patient CH30 in the main text.

  1. The caption of section 3.3 and 3.4 are the same. Please fix.

Reply: we apologize for the mistake. We fixed the error properly.

  1. GO enrichment results should be provided in supplemental materials.

Reply: we thank the reviewer for the precious suggestion. We added the GO enrichment results within the supplementary materials. Table S2 shows the enrichment results of the crossed-patient CTC set, while Table S3 provides results obtained through enrichment analysis of the entire case series.

Reviewer 2 Report

The authors present an exploratory paper on single CTC analysis in BC for CNA and SNV.

The paper is technically clearly written using established (commercial) techniques such as CellSearch, DEPAray, Ampli1 lowpass, Ampli1 Oncoseek for respectively CTC enrichment, isolation and CNA and SNV analysis.

The paper lacks novelty because the dataset is to limited(few patient, very limited follow up time points) and heterogeneous to correlate the results with therapy and/or response.

The discussion contains a lot of over interpretation, especially regarding SNV, and can be shortened.

The observation a group of CTCs that share common features may be of interested for further research, as the authors suggest, however these group already shows several in fig 4b and would likely be separated by PCA using a lager dataset. Comparing the SNV results of these 15 CTCs with the other CTS would be of interest.

Explain the quality issue for SNV analysis of CH28.

SNV analysis of CH46 and CH47 are missing in 3.2.

Only 32/55 CTC displayed a called variant in SNV analysis, CH32 only 4/15 (and these 4 CTCs al have 2 or more variants). Discuss this unexpectedly low frequency and compare to CNV profiles and literature.

Please, report the recovery rate for CTC isolation with DEPArray from the Cellsearch cartridge and single cell harvest recovery of DEPArray.

Paragraph heading of 3.3 and 3.4 are equal, please specify.

Indicate the (big difference between) number of CTCs for each time point in fig 2 and 3.

Are all Genomic copy number aberration profiles of CTCs representative for CTC showing increased aberrations compared to Lymphocytes as copy number variations are a hallmark of cancer. Can you conclude that based on the CNV profiles all identified CTCs are of tumor origin?

Report each copy number aberration profile as supplementary data.

Author Response

We thank the reviewer for revising our manuscript. We believe that that the reviewer’s comments significantly contributed to improve our manuscript. Please find below a point-by-point reply to the comments. 

The authors present an exploratory paper on single CTC analysis in BC for CNA and SNV.

The paper is technically clearly written using established (commercial) techniques such as CellSearch, DEPAray, Ampli1 lowpass, Ampli1 Oncoseek for respectively CTC enrichment, isolation and CNA and SNV analysis.

The paper lacks novelty because the dataset is to limited(few patient, very limited follow up time points) and heterogeneous to correlate the results with therapy and/or response.

Reply:  We agree with the reviewer with the issue concerning the limited number of patients included in our study. However, we are confident that this weakness is mitigated by high-quality single-cell data illustrated in this article, especially with whole-genome copy number aberration profiling, together with the high number of single cells analysed. We discuss this point in lines 345-348.

The discussion contains a lot of over interpretation, especially regarding SNV, and can be shortened.

Reply: we thank the reviewer for the kind suggestion, we shortened the discussion in particular the part concerning the SNV, as requested.

The observation a group of CTCs that share common features may be of interested for further research, as the authors suggest, however these group already shows several in fig 4b and would likely be separated by PCA using a lager dataset. Comparing the SNV results of these 15 CTCs with the other CTS would be of interest.

Reply: we thank the reviewer for this observation. The PCA separation on the entire dataset is reported in figure 4a where we added a circle to highlight the group of CTCs with common features and modified the figure caption accordingly. Concerning the comparation of SNV results in the 15 CTCs with common features, this investigation was originally included in our analyses. However, the SNV profile was available only for 9 out of 15 CTCs and did not provide any remarkable result, thus we did not report the results.

Explain the quality issue for SNV analysis of CH28.

Reply: we thank for the comment. We clarified the quality issues of CTCs from patients CH28 (lines 176-177).

SNV analysis of CH46 and CH47 are missing in 3.2.

Reply: we thank the reviewer for the suggestion. However, we were unable to include SNV analysis of CTCs from patients CH46 and CH47 since any cell resulted in sufficient quality at quality control (lines 177-178).

Only 32/55 CTC displayed a called variant in SNV analysis, CH32 only 4/15 (and these 4 CTCs al have 2 or more variants). Discuss this unexpectedly low frequency and compare to CNV profiles and literature.

Reply: we thank the reviewer for the valuable comment, and we agree with the suggestion. We implemented the discussion as requested in lines 365-368. The investigated panel includes hotspot mutations occurring in 60 oncology-relevant genes. Missing variant calling may be due to the fact that CTCs have mutations occurring on genes or exons not included in the panel that would require the increase of the number of genes in the analysis.

Please, report the recovery rate for CTC isolation with DEPArray from the Cellsearch cartridge and single cell harvest recovery of DEPArray.

Reply: we thank the reviewer for this comment. At this time, we are not able to discuss this data in this paper, as the CellSearch enumeration per patient will be included in another article. In brief, we found that at DEPArray analysis the individuation rate was nearly 30-40%. Moreover, the conditions were optimal and we did not observe any harvesting issue for recovery at DEPArray.

Paragraph heading of 3.3 and 3.4 are equal, please specify.

Reply: we apologize for the mistake. We fixed the error properly.

Indicate the (big difference between) number of CTCs for each time point in fig 2 and 3.

Reply: we are sorry for the missing information. We indicated the number of CTCs included in the longitudinal analysis in lines 275-276 and 284-285.

Are all Genomic copy number aberration profiles of CTCs representative for CTC showing increased aberrations compared to Lymphocytes as copy number variations are a hallmark of cancer. Can you conclude that based on the CNV profiles all identified CTCs are of tumor origin?

Reply: we thank the reviewer for the comment. In our analysis, we did not find any significant aberration occurring on lymphocytes. As a consequence, our analysis confirmed the malignant nature of the aberrations found in CTCs. We edited the text in lines 227-229 to include this sentence.

Report each copy number aberration profile as supplementary data.

Reply: We thank the reviewer for the suggestion. We prepared pdf files containing plots of copy number aberration profiles of each analysed CTC. However, in our opinion, the main focus of the manuscript concerns in the elaboration of these raw data, and we think that including all the profiles may be redundant. We propose to create a supplementary file including only plots of the most representative CTCs for each patient (Please see the doc file Supplementary figure S4 (candidate alternative to zip file Supplementary Material 2)).

Round 2

Reviewer 2 Report

We thank the authors for the adjustments they made to the manuscript. It has made the manuscript more readable and clear by providing all the necessary information.